# Human Regulatory T Cells: Understanding the Role of Tregs in Select Autoimmune Skin Diseases and Post-Transplant Nonmelanoma Skin Cancers

**DOI:** 10.3390/ijms24021527

**Published:** 2023-01-12

**Authors:** Nicole Chizara Oparaugo, Kelsey Ouyang, Nam Phuong N. Nguyen, Amanda M. Nelson, George W. Agak

**Affiliations:** 1David Geffen School of Medicine at UCLA, Los Angeles, CA 90095, USA; 2Division of Dermatology, David Geffen School of Medicine at UCLA, Los Angeles, CA 90095, USA; 3Cleveland Clinic Lerner College of Medicine, Case Western Reserve University, Cleveland, OH 44195, USA; 4Morsani College of Medicine, University of South Florida, Tampa, FL 33620, USA; 5Department of Dermatology, Penn State University College of Medicine, Hershey, PA 17033, USA

**Keywords:** T-regulatory cells, autoimmunity, T effector cells, skin, inflammation, homeostasis, immune suppression, transplantation, cancer

## Abstract

Regulatory T cells (Tregs) play an important role in maintaining immune tolerance and homeostasis by modulating how the immune system is activated. Several studies have documented the critical role of Tregs in suppressing the functions of effector T cells and antigen-presenting cells. Under certain conditions, Tregs can lose their suppressive capability, leading to a compromised immune system. For example, mutations in the Treg transcription factor, *Forkhead box P3* (FOXP3), can drive the development of autoimmune diseases in multiple organs within the body. Furthermore, mutations leading to a reduction in the numbers of Tregs or a change in their function facilitate autoimmunity, whereas an overabundance can inhibit anti-tumor and anti-pathogen immunity. This review discusses the characteristics of Tregs and their mechanism of action in select autoimmune skin diseases, transplantation, and skin cancer. We also examine the potential of Tregs-based cellular therapies in autoimmunity.

## 1. Introduction

Immune system homeostasis is tightly regulated by the proper functioning of regulatory T cells (Tregs), a T cell subpopulation that utilizes various suppressive mechanisms to modulate the activity of other immune cells [1]. The interaction between Tregs and other cells in the immune system is a necessary step in the maintenance of self-tolerance. From a functional perspective, Tregs are mainly responsible for suppressing the activation, proliferation, and cytokine production of CD4^+^ and CD8^+^ T cells and are also believed to suppress B cells and dendritic cells [2,3,4]. A notable genetic marker of Tregs is the transcriptional factor, Forkhead box P3 (FOXP3), which serves as a master regulator for Treg development and function [1,5]. A missense loss-of-function mutation within the FOXP3 locus can result in self-reactive lymphocytes that can lead to the development of severe autoimmunity in scurfy mice or cause a rare, but severe, disease IPEX (immune dysregulation, polyendocrinopathy, enteropathy, X-linked) syndrome in humans [1,2,3]. Therefore, the importance of Tregs is evident in that these cells are crucial in orchestrating immune suppression and helping to prevent autoimmune disease.

The functional stability of Tregs is required to control inflammation. Using FOXP3 fate reporter mice, Rubtsov et al. demonstrated that Tregs are a highly stable lineage [6]. However, in certain disease states, Treg lineage instability has been reported. In such instances, previously FOXP3-positive Tregs lose FOXP3 expression and demonstrate an effector T cell (Teff) phenotype [7,8,9]. The factors and tissue-specific cues that cause a total loss of Treg identity and partial shift towards Teff phenotype, yet still maintain the suppressive function, are yet to be elucidated. One proposed explanation for the loss of Treg identity is that a proportion of FOXP3^+^ Tregs only transiently express FOXP3 and lack the complete epigenetic Treg-cell program [10,11]. 

In healthy humans, circulating Tregs represent a highly heterogeneous population of phenotypes and gene expression profiles, with some FOXP3^+^ cells demonstrating close similarity to conventional T cells (T_conv_). This means that FOXP3 is a highly specific, but not an absolute, Treg marker. As such, this indicates the importance of understanding the heterogeneity of FOXP3^+^ T cells in different clinical settings, including autoimmune skin diseases, transplantation, and cancer. T_conv_ broadly requires IL-7, whereas Tregs express the IL-2 receptor α-chain (CD25) and are dependent on IL-2 [12]. In all, Tregs circulating in human blood can be divided into different fractions and isolated using markers based on CD25^+^ FOXP3^+^ expression and several other markers that are beyond the scope of this review [13,14,15,16]. We first describe Tregs classification, followed by the proposed mechanisms of action and function in select disease settings. Lastly, we discuss the potential of Tregs-based therapies in autoimmunity. 

## 2. Classification of Tregs 

In response to environmental antigens and cues, Tregs either develop from autoreactive thymocytes in the thymus or naïve CD4^+^ T cells in the periphery [17]. Thymic-derived Tregs (tTregs) mature directly from CD4 and CD8 double-positive T cells within the thymus and are sometimes referred to as naturally occurring Tregs (nTregs). In this pathway, thymic T cells recognize self-antigen–MHC complexes expressed on thymic epithelial cells with relatively high avidity, leading to the development of a T cell receptor (TCR) repertoire with a self-bias necessary for autoimmune prevention [18,19,20]. On the other hand, peripheral Tregs (pTregs) develop from differentiated naïve T cells in the periphery upon stimulation with IL-2 and transforming growth factor-β (TGF-β) in vivo. The term inducible Tregs (iTregs) refers to Tregs generated with IL-2 and TGF-β in vitro. Both tTregs and pTregs differ in ontogeny, regulation, and function, ensuring their complementary role in maintaining immune tolerance and homeostasis [21,22]. TGF-β, which is predominately secreted by CD103^+^ dendritic cells (DCs), is essential for the generation of both tTregs and pTregs [23]. Several commensal bacteria can also induce TGF-β secretion by DCs [24,25]. Overall, tTregs and pTregs can regulate inflammation in different tissues throughout the body. 

## 3. Tissue-Specific Tregs

Tregs are highly complex immune cells that carry out various functions specific to the peripheral tissues in which they reside [26]. Tissue-specific Tregs are found within the gastrointestinal tract (GI), visceral adipose tissue (VAT), and the skin, where they act as specialized suppressors of inflammation [27,28]. For example, within the GI tract IL-33-responsive GATA3^+^ Helios^+^ colonic tTregs ameliorate tissue damage during colitis [29]. In contrast, GATA3^+^ Helios^+^ colonic pTreg cells and RORγt^+^ Helios^−^ Tregs are induced by intestinal microbiota, and the loss of RORγt expression results in severe intestinal inflammation [30,31,32]. Additionally, within the epithelial layer of the small intestines, food antigens have been shown to induce the RORγt^−^ Helios^−^ Tregs subset that prevents allergic responses to food antigens [33]. Tregs are also present in healthy skeletal muscle and increase in numbers following muscle injury. In this case, the increased frequency of Tregs not only helps to suppress inflammation, as would be expected, but also produce factors, such as amphiregulin, which enhance muscle regeneration and repair [34]. Tregs in VAT sites express the transcriptional factor, PPARγ (peroxisome proliferator-activated receptor gamma), which is associated with the differentiation of adipocytes. VAT-residing Tregs control inflammatory states of adipose tissue, and their depletion abrogates metabolic parameters, such as insulin sensitivity [16,28]. 

Skin-resident Tregs have features of memory Tregs (mTregs) due to their ability to maintain homeostasis and residency within the cutaneous environment long after initial antigen exposure [35,36]. Phenotypically, human cutaneous Tregs express classic memory T cell markers, such as CD45RO [36]. In steady-state conditions, mTregs localized in the hair follicle stem cell niche are non-migratory and nearly unresponsive. These skin-resident Tregs produce Jagged1, a Notch ligand that is associated with hair regrowth and the differentiation of stem cells within the hair follicle [37]. In inflammatory microenvironments, mTregs in psoriatic lesions are rapidly proliferative, secreting low levels of proinflammatory cytokines, such as IL-17. In addition to attenuating inflammatory responses, Tregs in the skin aid in skin repair [38], maintain tolerance with commensal skin microbes, and assist in hair follicle regeneration [37,39].

## 4. Mechanism of Action

Tregs are a functionally heterogeneous population that can suppress immune responses through various versatile and complementary mechanisms [40]. Often this suppression is highly specific to certain types of immune responses. Although the molecular mechanisms by which Tregs exert their suppressor activity are not clearly defined, we discuss some proposed mechanisms of immune suppression (Figure 1). 

### 4.1. Starving T Cells of IL-2

IL-2 secreted by thymic cells, DCs, and activated T cells is consumed in an autocrine/paracrine manner by cells that harbor IL-2R. Tregs express IL-2R and are physiologically primed to proliferate constitutively in the presence of IL-2. In fact, proper maturation of Tregs in the thymus depends heavily on IL-2R signaling [47,48,49]. Murine studies have shown that mice lacking components of the IL-2/IL-2R signaling axis develop autoimmune diseases [50,51], suggesting that IL-2 is critical in generating functional Tregs (Figure 1). The expression of the high affinity IL-2R allows Tregs to lower the IL-2 concentrations within the microenvironment, thereby decreasing the proliferative signal available to other T cell subsets [52]. 

### 4.2. Induction of T Cell Apoptosis

Tregs can inhibit Teff activity by inducing apoptosis via the release of cytotoxic molecules, perforin, and granzyme B (GzmB) [53]. Gondek et al. reported less effective Treg activity in GzmB-deficient mice, demonstrating the importance of GzmB in Treg-mediated suppression [54]. In addition to GzmB, the expression of TNF-related apoptosis-induced ligand (TRAIL), a suppressor that acts via interaction with death receptor 5 on CD4^+^ T and other effector cells, is pivotal. TRAIL activation results in caspase-8-mediated apoptosis [55,56]. 

### 4.3. Production of Inhibitory Cytokines

Tregs secrete high amounts of cytokines that have immunosuppressive actions, such as TGF-β, IL-10, and IL-35 (Figure 1). These cytokines have non-specific suppressive activity and can target effector B and T cells [57,58,59]. IL-10 and TGF-β can inhibit antigen presentation by DCs and enable the induction of pTreg populations, such as Th3 and Tr1 cells. Such Tregs have been observed at sites of chronic inflammation and in transplanted tissues [60,61,62]. Moreover, Tregs can induce other cell types to express IL-10 [63].

### 4.4. Inhibiting Immunostimulatory Signals and Metabolic Activity of APCs 

Treg recognition of antigens presented via MHC-II molecules can lead to the generation of non-functional antigen-presenting cells (APCs) that cannot present antigens. The methods of Treg suppression are diverse and can include the binding of costimulatory molecules CD80/86 on APCs by CTLA-4 [64], removal of Ag-MHC-II complexes from the APC surface through trans-endocytosis [65], decreased indoleamine 2,3-deoxygenase (IDO) expression on the APC surface leading to reduced tryptophan levels, and the subsequent loss of proliferative capacity [66]. In this way, IDO acts as a rate-limiting enzyme that aids in sustaining Treg-mediated immune tolerance. When expressed on DCs, the enzyme drives the differentiation of naïve CD4^+^ T cells towards a FOXP3^+^ phenotype, enhances the Treg suppression of Teffs [67,68], and prevents the conversion (“reprogramming”) of Tregs into pro-inflammatory cells [69]. These processes disrupt the ability of APCs to process and present antigens, subsequently leading to T cell anergy. In addition, because of its immunoregulatory capacity, it has been proposed that IDO expression can suppress T cell responses and promote immune tolerance in mammalian pregnancy, autoimmunity, and allergic inflammation [70].

The disruption of Treg activity augments critical memory processes and immune system homeostasis. Understanding the role of cutaneous mTregs may open the door to tremendous breakthroughs in therapeutics. In this review, we focus on the role of Tregs in select autoimmune skin conditions and post-transplant cancers (Table 1). 

## 5. Role of Tregs in Select Autoimmune Skin Diseases

### 5.1. Psoriasis

Cutaneous psoriasis is a prevalent disease that affects roughly 2–5% of the world’s population [98]. Patients with this condition may report debilitating symptoms that impact their physical and mental well-being. Psoriatic lesion sites contain an infiltration of immune cells and abnormal proliferation of keratinocytes within the dermis and epidermis [99]. 

Though not fully understood, the pathogenesis of psoriasis is complex and multifactorial, influenced by both the environment and genetic factors [100,101]. Environmental triggers include tobacco, infections, stress, and physical trauma [71,102]. Twin studies conducted by Brandrup et al. provide additional insight into the genetic component of psoriasis. Here, monozygotic twins were found to have a 40% concordance rate of psoriasis compared to a dizygotic twin concordance rate of 10% [103]. Furthermore, 35% of psoriasis patients reported a positive family history of psoriasis [104,105]. HLA genetic studies have identified increased MHC class I antigens in individuals with psoriasis [106]. A greater understanding of psoriatic triggers is needed and as our understanding grows, so do the opportunities for developing novel treatment options.

In psoriasis, Tregs lose their ability to effectively suppress the excessive expansion of Th17 cells [107]. The increased IL-17 levels are proposed to intensify T bet expression and IFN-γ production by further downregulating FOXP3 and TGF-β expression [108]. Because FOXP3 is vital to Treg development in the thymus, such an environment results in poorly developed Tregs, thereby exacerbating inflammation rather than suppression [109,110,111,112]. Rather than dampen these immune responses via FOXP3 repression of RORγt, Tregs that have undergone plasticity are reported to worsen inflammation by releasing IL-17, IFN-γ, and TNF-α via the phosphorylation of STAT3 [112]. In vitro, STAT3 phosphorylation occurs in the presence of IL-6, IL-21, and IL-23. These cytokines likely play a role in Treg dysfunction and transformation into Th1/Th17 cells [112,113,114]. This unchecked inflammatory environment where Tregs have lost their immunosuppressive ability and gained expression of inflammatory cytokines is critical for the progression of psoriasis. 

### 5.2. Vitiligo

Vitiligo is an autoimmune disease characterized by the destruction of pigment-producing melanocytes by CD8^+^ T cells. Vitiligo affects approximately 0.5–2% of people worldwide [115,116]. The disease presents as white patches on the skin, hair, and mucous membranes. The development of vitiligo is attributed to oxidative stress that leads to the activation of the immune system and the attack of melanocytes [117]. Stress proteins, heat shock proteins (Hsp) 70i, and several chemokines, including CXCL9, CXCL10, and CXCL11, play a crucial role in the recruitment of cytotoxic CD8^+^ T cells to the skin. The extent of melanocyte destruction has been directly correlated with the amount of CD8^+^ T cell infiltration [117,118,119,120,121,122]. In addition, CD8^+^ T cells with cytotoxic activity against autologous melanocytes localize at the dermal/epidermal junction [121,123,124]. This loss of self-tolerance suggests that Tregs may be involved in the pathogenesis of vitiligo. 

Discrepancies have been reported in the number of Tregs present in vitiligo patients and healthy controls. Depending on the study and study population, patients with non-segmental vitiligo were found to have decreased, unaltered, and increased levels of circulating Tregs [125,126,127,128,129,130,131]. Further inspection of these studies revealed differences in the antibody markers used to quantify Tregs. Using FOXP3^+^ or CD25^+^ alone as a marker for Tregs in human skin provides limited information, since activated CD4^+^ T cells can also express these two markers [128,132]. The small sample size and variations of Treg markers used by these studies could explain the variation in the observed results of circulating Tregs in generalized, non-segmental vitiligo. Identifying a specific marker for Tregs will help resolve this issue, and in doing so, our ability to monitor the development of vitiligo will improve.

Chemokines are a family of small, highly conserved cytokines that mediate critical biological processes, such as chemotaxis, hematopoiesis, and angiogenesis. Several subfamilies of chemokines (e.g., CXC, CC, C, and CX3C) have been defined by the positions of sequentially conserved cysteine residues [133]. Because of their significant involvement in various pathologies, chemokines and their receptors have been the focus of therapeutic discovery for clinical investigations in vitiligo. Several studies demonstrate reduced levels of CCL5/CCR4, CCL22, CCL21, and CCR6 in vitiligo skin, which could explain the failure of circulating Tregs to localize to the skin [126,129,134,135,136,137], thus suggesting that the modulating expression of these chemokines may be potential therapeutic targets in vitiligo. Therefore, it is envisaged that after modulation, these chemokines may re-establish proper immune regulation and self-tolerance by increasing the frequency of Treg migration and skin homing into vitiligo lesions [116,136].

Some Tregs respond to specific environmental cues, such as nutrients, metabolites, and cytokines (Figure 1). In such cases, Treg stability may be altered, leading to plasticity that impacts their suppressive functions [41]. Chen et al. demonstrated that normal Tregs could transition into Th1-like T-bet^+^IFN-γ^+^ Tregs in vitiligo patients, and serum from vitiligo patients caused normal Tregs from healthy control subjects to transition into Th1-like Tregs [138,139]. These Th1-like Tregs had attenuated suppressive activity against CD8^+^ T cells, which was consistent with previous findings [127,130]. Recently, studies have demonstrated that Th1-like Tregs promote tissue-resident memory CD8^+^ T cells, which have been shown to play a role in vitiligo relapse [140,141,142,143]. Understanding the mechanisms that regulate Treg conversion into Th1-like cells will provide new insights into immune homeostasis and disease pathogenesis, with important therapeutic implications for vitiligo patients.

### 5.3. Systemic Sclerosis 

Affecting approximately 1 in every 10,000 individuals globally, systemic sclerosis (SSc) is a rare autoimmune disease that results in the fibrosis of connective tissues and various organs [144]. The disease commonly begins with Raynaud’s phenomenon, followed by gastro-esophageal reflux [145,146]. Although not yet fully understood, reports suggest that the age of onset may influence prognosis and disease severity as SSc remains a rheumatic disease with the highest mortality rate [147,148]. 

SSc is multifactorial driven by both genetic [149] and environmental [150] influence. The pathology of this autoimmune disorder is characterized by a dysregulated immune system, consisting of a Th17/Treg imbalance as reported by several studies that noted decreased levels of Tregs and increased levels of Th17 cells in both peripheral blood and skin lesions of SSc patients [89,151,152,153]. The accumulation of circulating pTregs and IL-17-producing T cells may be driven by Treg plasticity towards the Th17 phenotype, leading to the Th17/Treg imbalance. Although the mechanism is largely still unclear, available data suggests a correlation between increases in Th17-associated cytokines (IL-17, IL-21, and IL-22) and SSc severity [89,154]. Some studies propose that Th17 cells induce collagen secretion to promote fibrosis [89,155], while others suggest that Th17 cells promote inflammation and fibrosis [90,156,157,158]. Further, reduced levels of IL-10 present in the serum of SSc patients may be secondary to impaired Treg activity [159,160]. Taken together, Treg plasticity towards Th17-like expression and suppressive function implies an essential role for Tregs in SSc. 

IL-4, IL-13, and IL-33 may contribute to the inflammatory phase, but the cellular origin of these cytokines in SSc is unclear [161,162,163]. Evidence suggests that IL-33 can induce increased numbers of Th2-like Tregs cells in the skin of patients with SSc [91,164]. Although the mechanism is not yet fully understood, Slobodin et al. reported a positive correlation between CD4^+^CD25^bright^Foxp3^+^ Tregs and SSc disease severity [165]. Further research into the mechanism by which IL-33 regulates Treg transdifferentiation and plasticity are, therefore, warranted.

Additional studies observed clinical improvement in SSc patients treated with Fresolimumab, an antibody that targets TGF-β-producing Tregs [166]. In aggregate, studies support the premise that Tregs fail to produce inhibitory cytokines or suppress Teffs in SSc, but the underlying mechanisms of how this occurs are unclear. Future work needs to address how the failure of Treg-mediated immune suppression contributes to SSc.

## 6. The Role of Tregs in Transplantation and Skin Cancer

Following transplantation, host immune defenses can recognize allografts as foreign and mount an immune attack against the graft [167]. The excess secretion of cytokines, chemokines, and other effector molecules further amplifies alloimmune responses, resulting in graft rejection [168]. To promote tolerance, immunosuppression regimens have been integrated into transplant care and management, resulting in drastic improvements in postoperative outcomes [169,170]. Immunosuppressants help re-educate the immune system during transplantation by suppressing immune attacks against the foreign graft [171]. The suppression of the immune system, while essential for transplant success, increases the likelihood of various complications [172,173]. By inhibiting Teffs and enhancing Tregs activation, immunosuppressive therapies can dampen the immune response to various pathogens and tumor cells [174]. 

The ratio of Tregs versus CD8^+^ T cells affects the success of transplantation, with higher Treg/CD8^+^ T cell ratios associated with improved transplant tolerance [12,175,176]. Given the role of Tregs in maintaining tolerance to self and foreign antigens, extensive work has been directed toward the utilization of Treg-based cellular therapies to optimize transplant survival. For example, multiple studies have considered the potential use of pharmacologically induced tolerogenic Tregs in post-transplant care [177]. Interestingly, the same mechanisms by which the increased ratio of Tregs can dampen immune responses are also the ones that may contribute to worse clinical outcomes following transplantation, such as the increased risk of cancer [178]. 

Immunosuppressed patients are at an increased risk for developing numerous malignancies, with skin cancer accounting for nearly 40% of malignancies in organ transplant recipients [178]. Prior studies have identified an increased presence of Tregs surrounding the tumor and lymphoid tissues that drain the tumor [179,180]. Furthermore, an increased Treg/CD8^+^ T cell ratio has been suggested to increase the likelihood of tumor evasion [181,182,183]. By secreting a wide array of anti-inflammatory cytokines, including IL-10, IL-35, and TGF-β, Tregs can prevent pro-inflammatory and anti-tumor responses [176,184]. Consequently, increased T cells predict worse prognoses for various cancer patients [180,185]. Below, we review the two most commonly reported skin cancers following transplantation, namely squamous cell carcinoma (SCC) and basal cell carcinoma (BCC), focusing specifically on the involvement of Tregs. 

### 6.1. Squamous Cell Carcinoma

Cutaneous squamous cell carcinoma (cSCC) accounts for the highest proportion of post-transplant skin cancers [186]. Immunosuppression not only increases the risk of cSCC development but also contributes to worsened cSCC severity. Observations of immunosuppressed individuals demonstrate a more aggressive cSCC phenotype when compared to immunocompetent patients [187]. Transplant patients have an estimated 65 to 108 times higher risk of cSCC development than the general population [188]. Additionally, organ transplant recipients with cSCC have a 60 to 250 times greater risk of mortality from SCCs than immunocompetent individuals [189,190,191,192,193]. 

The high numbers of Tregs are associated with the development of cSCC in renal transplant patients [178]. In a follow-up study, analysis of the Treg-specific demethylated regions confirmed that kidney transplant recipients with a prior history of developing a cSCC had a higher proportion of Tregs than cytotoxic T cells in the immune microenvironment [194]. Using single cell TCR sequencing, Frazzette et al. demonstrated that immunosuppressed patients had a lower proportions of cytotoxic T cells in comparison to immunocompetent patients [195]. In addition, a distinct population of CD8^+^FOXP3^+^ T cells that had not previously been observed in cSCC were also described [195]. These unique Treg subpopulations expressed cytotoxic molecules, such as perforin [196,197]. Further studies are needed to elucidate the role of these CD8^+^ T cells as they may offer a unique novel immunotherapeutic avenue for the treatment of cSCC in transplant recipients.

### 6.2. Basal Cell Carcinoma

BCC is the second most common malignancy in solid organ transplant recipients (SOTRs) with SOTRs having a 10-fold higher risk compared to immunocompetent individuals [191]. Interestingly, a clinicopathologic study of 176 cases revealed that BCCs in SOTRs have distinct clinical characteristics compared to immunocompetent patients [198]. For example, even though BCCs in immunocompetent individuals and organ transplant recipients are typically seen on the head and neck, there is a higher percentage of SOTRs with BCCs that are localized in sun-protected sites, such as genitalia and the axilla, which are absent in immunocompetent individuals [198]. 

A closer histological examination of BCCs indicated that the peritumoral inflammatory cell infiltrates were significantly lower in SOTRs compared immunocompetent patients. Additionally, peritumoral skin showed a higher concentration of Tregs, while normal, non-UV-exposed buttock skin lacked Treg expression [198]. In a study by Omland et al., Tregs expressing CCL17, CCL18, and CCL22 were found accumulated in BCC tumors [199]. These chemokines are involved in the recruitment of Tregs in solid cancers [199]. Additionally, genome-wide association analyses (GWA) and functional interaction network analyses have revealed the enrichment of risk variants that function in an immunosuppressive regulatory network, which can impair immune surveillance and effective antitumor immunity [200]. In addition, the GWA data also revealed a global enrichment of genes linked to Treg-cell biology, underlining the importance of Tregs for BCC development. 

## 7. Regulatory T Cell Therapies

Despite the many complications, immunosuppressive drugs remain the cornerstone of transplant medicine and many autoimmune conditions [201,202]. Non-specific actions of these drugs lead to unwanted side effects, such as increased risk of infection and malignancy [203,204,205]. Technological advances have helped define and characterize Tregs, increasing our understanding of the balance between Treg plasticity and Treg instability. In recent years, advances in cellular Treg-based therapies have sought to harness Treg’s unique ability to induce immune tolerance. Here, we summarize several promising Treg-targeted therapies and approaches that are currently under trial. 

### 7.1. IL-2 Based Approaches: Low Dose IL-2 and IL-2 Complexes 

IL-2 was first described by its ability to mediate T cell growth [48] and was later found to be involved in several other immune pathways involving natural killer cells and CD8^+^ T cell cytolytic activity [206], CD4^+^ T cell differentiation, and Treg expansion [207]. The role of IL-2 has proven to be critical in Treg proliferation and survival [48], and modifications to the amount of IL-2 present in the microenvironment can alter the Treg:Teff ratio, impacting immune tolerance. Low dose IL-2 approaches are based on the expression of the IL-2 receptor on Treg cells and allow for the preferential expansion of Tregs with low levels of IL-2 [208,209,210]. Further, anti-IL-2 monoclonal antibodies (mAb) can be designed to selectively expand Treg expression [211] and IL-2 fusion proteins (IL-2 bound to the Fc portion of IgG) work by slowing the IL-2 excretion rate [212]. In patients with alopecia areata, low-dose IL-2 treatment expanded Treg cells in the blood and hair follicles, leading to improved hair growth [213]. Klatzmann et al. further demonstrated the potential of low dose IL-2 strategy in the treatment of several other autoimmune diseases, including psoriasis [209]. These outcomes certainly highlight the potential of IL-2 based approaches in targeting autoimmune and inflammatory conditions [214,215].

Early findings from case reports showed promising results of IL-2 agents for psoriasis. For example, use of basiliximab, a chimeric IL-2 mAb, was effective for the treatment of severe psoriasis [216,217]. Additionally, a clinical trial demonstrated a reduction in the psoriasis area and severity index (PASI) by 30% at 8 weeks with use of daclizumab, a humanized IgG1 mAb that blocks the IL-2 receptor by binding to CD25 on T cells [218]. Despite early positive results, α-IL-2 agents have fallen out of favor for treatment of psoriasis due to relatively high toxicity and intermediate treatment response compared to other biologics for psoriasis. 

However, the development of α-IL-2 therapeutics remains of interest for other conditions [219] as findings from a recent 24-week study by Yu et al. suggest that IL-2 may be used for SSc without obvious adverse effects. The group confirmed previous findings showing that SSc typically has an imbalance of T cells and showed that low-dose IL-2 therapy can restore the balance of the Th17 to Treg cell ratio, leading to reduced disease activity in SSc patients [220]. Furthermore, many centers have incorporated IL-2 antagonists in post-transplant immunosuppressive regimens [221]. Still, the results are unclear as to whether the use of IL-2 receptor antagonists are associated with improved post-transplant outcomes [221,222]. For example, a meta-analysis by Ali et al., demonstrated that IL-2R antibody induction therapy did not improve the rate of rejection or graft survival for renal transplant recipients on tacrolimus maintenance therapy. Therefore, further investigations and development of randomized controlled trials to examine the use of IL-2 immunosuppressive therapies are needed.

### 7.2. Adoptive Treg Therapy: Polyclonal and Antigen-Specific Tregs 

When compared to conventional immunosuppressive agents, ex vivo generated Tregs have been associated with fewer adverse events when used in the setting of autoimmune diseases and transplant rejection [223,224,225,226]. Trzonkowski et al. were the first to use in vitro expanded Tregs in humans and reported symptomatic relief in patients with graft versus host disease (GvHD). In 2004, this treatment was later used in type 1 diabetes, where it demonstrated efficacy, minimized serious side effects, and showed the reversal of the disease [227]. In more recent years, studies have demonstrated the therapeutic potential of adoptive Treg therapy in cutaneous autoimmune conditions, such as vitiligo and SSc. In 2014, Chatterjee et al. reported disease remission in vitiligo mice models that received adoptive transfer of purified Tregs [228] and phase l/ll clinical trials are currently underway for the use of adoptive Treg therapy in the treatment of SSc (NCT05214014).

The challenge with polyclonal is the large quantity of Tregs needed to reach therapeutic levels. An alternative to polyclonal Tregs is the use of antigen-specific Tregs, a form of adoptive Treg therapy that offers a more targeted solution and requires the injection of fewer cells. The classic approach for designing antigen-specific Tregs utilizes either APCs and distinct antigens or engineered T cell receptors (TCRs) [229]. Although both may be reasonable options, limitations exist in the rate of Treg expansion when using APCs and MHC restrictions when engineering Tregs with TCRs. Thus, chimeric antigen receptor (CAR) technology is a preferred method for engineering antigen-specific Tregs that are non-MHC-restricted [230]. CARs bind specified antigens with high affinity using their extracellular antigen-binding domain comprised of a single chain variable fragment (scFv) [230]. The concept of CARs was first described over 25 years ago by Gross et al. and has been improved upon over time [231]. Today, the technique is being studied in various autoimmune connective tissue conditions [232], holding promise for the potential treatment of SSc [233]. Furthermore, the use of CAR-T-cell (CAR-T) therapy in BCC resulted in partial tumor regression [234]. A number of case reports have also documented the use of CAR-T therapy in post-transplant lymphoproliferative disorders in SOTRs [235,236,237,238,239,240,241,242,243]. As our understanding of CAR-T therapy improves, its diverse applications continue to grow.

## 8. Future Directions

The profound effects that Treg cell disruption has on immune system homeostasis has led to novel clinical investigations. Current human clinical trials are examining the effect of depleting Tregs as a cancer therapy. Furthermore, investigators are exploring the converse: expanding and transplanting Tregs for treating autoimmune diseases. However, despite significant progress, gaps in our understanding of Treg function within the skin and other organs still exist. Recent advances in single cell RNA sequencing, notably the possibility to analyze the Treg-transcriptome from different sites (tumor, diverse tissues, circulation, and the tumor microenvironment) at the single cell level, will provide unprecedented detail on the heterogeneity of these cells that may be exploited in future therapies. Overall, complete elucidation of the mechanism driving Treg plasticity would allow for the successful development of targeted Treg-based therapies. 

## Figures and Tables

**Figure 1 ijms-24-01527-f001:**
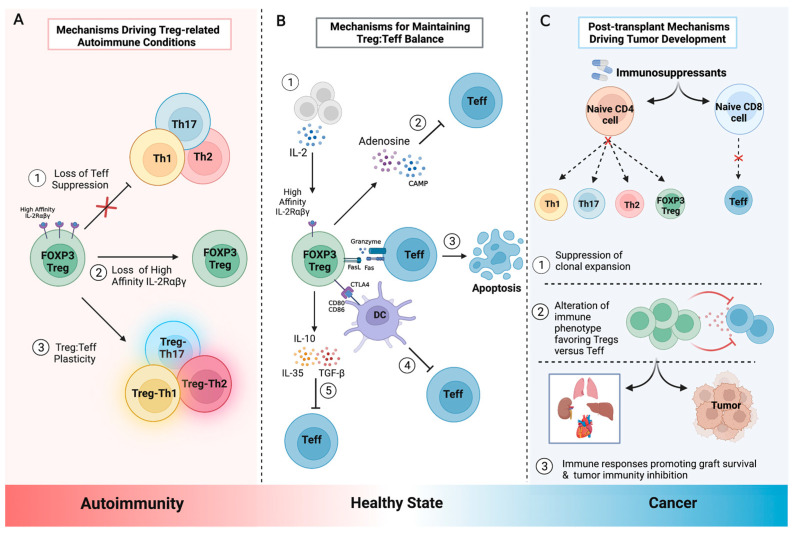
Tregs homeostasis and dysregulation. Tregs interact with effector T cells (Teffs) to modulate the immune responses and maintain self-tolerance. Tregs act by suppressing Teffs through various mechanisms that, when disrupted, can lead to several autoimmune conditions and malignancies. (**A**) Mechanisms that drive autoimmunity can occur via: (1) Loss of the ability of Tregs to inhibit the activity of Teffs, leading to an overactivation of the immune system. (2) Loss of high affinity IL-2R on the Treg cell surface leads to increased levels of IL-2 in the environment that is available for other T cell subsets to utilize for their proliferation. (3) Treg plasticity leads to an unstable phenotype that impacts their suppressive functions [41]. Notably, Treg plasticity towards Teffs activity can enhance inflammation when cytokines, such as IL-17, IFN-γ, and TNF-α, are released into the microenvironment. (**B**) Mechanisms that promote a healthy state involve: (1) Treg sequestration of IL-2 by high affinity IL-2Rαβγ, which decreases IL-2 availability for Teffs, indirectly inhibiting their survival. (2) Treg release of adenosine and cAMP impairs Teff metabolism and promotes homeostasis [42]. (3) Teff apoptosis can occur secondary to perforin and granzyme release by Tregs via FasL–Fas interactions. (4) Tregs have the ability to obstruct co-stimulation on APCs through CTLA-4, preventing Teff binding via CD28. (5) Treg release of anti-inflammatory cytokines IL-10, TGF-β, and IL-35 suppresses Teff activity. (**C**) Post-transplant mechanisms driving tumor development. Although it is still unclear how immunosuppressive therapies affect immune subtypes, evidence suggests that immunosuppressants affect clonal expansion and immune cell functionality by various mechanisms, such as decreased production of IL-2 and IFN-γ [43,44,45]. Immunosuppressants that favor increased ratio of Tregs versus Teffs have been associated with decreased incidence of graft rejection [46]. Conversely, an immunosuppressed environment can promote decreased immune surveillance and antitumoral responses, such as the inhibition of cytolytic responses against the tumor by Teffs, leading to tumor development [43]. Created with Biorender.com.

**Table 1 ijms-24-01527-t001:** Table summarizes the key effector cells, cytokines, reported and proposed treatments, and risk factors of select cutaneous autoimmune diseases. IL, interleukin; Th, T helper; Treg, T regulatory; TNF-α, tumor necrosis factor alpha; IFN-γ, interferon gamma; PDE-5 = phosphodiesterase 5; JAK, Janus kinase; and NB-UVB, narrowband ultraviolet B.2.3.

Disease	Risk Factors	Teff Cells	Cytokines	Treatments	References
Psoriasis	Obesity, infection, trauma	Th1, Treg-Th17	IL-2, IL-17, IL-22, IL-23, IL-26, TNF-α, IFN-γ	Anti-TNF-α inhibitors, T-cell-targeted therapies	[71,72,73,74,75,76,77,78]
Vitiligo	Genetics, trauma	Treg-Th1	IL-1β, IL-6, IL-15, IL-22, IL-33, TNF-α, IFN-γ	JAK inhibitor, NB-UVB therapy, anti-TNF-α inhibitors	[79,80,81,82,83,84,85,86,87]
Systemic sclerosis	Genetics, silica, solvents, heavy metal	Treg-Th2, Treg-Th17	IL-4, IL-13, IL-17, IL-21, IL-22, IL-33	Low-dose IL-2 therapy, PDE-5 inhibitors, calcium channel blocker, bosentan	[88,89,90,91,92,93,94,95,96,97]

## Data Availability

Not applicable.

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
