# Peer review of "Human Regulatory T Cells: Understanding the Role of Tregs in Select Autoimmune Skin Diseases and Post-Transplant Nonmelanoma Skin Cancers"

_ijms, 2023, doi:10.3390/ijms24021527_

Round 1

Reviewer 1 Report

This review manuscript comprehensively summarized the knowledge about Treg cells including the Treg ontogeny and mechanism of functions. Emphatically discussed the dysregulation of Treg cells in skin-associated autoimmune diseases and post-transplantation cancers. The structure and language of the manuscript are well-organized and easy to follow to the audience. I have a few minor suggestions as follows.   1. For Figure 1A the title mentions "Treg autoimmunity" which is unclear about the meaning. 2. There are some typos in the figure legend about the numbers of each mechanism. Please correct. 3. For section 4.2, the title is "inducing T cell apoptosis" but the second paragraph is not about T cell apoptosis, maybe the authors can add more detail about how the PD-1 pathway affects T cell apoptosis or change the title. 4. For section 4.4 in line 193 is there a sentence about how Treg down-regulate IDO on APC missing? 5. In line 276 there are some typos.

Reviewer 2 Report

This is an excellent review of T-cells in specific autoimmune conditions. Information has been pulled from numerous publications that are appropriately cited. 

One suggestion: If at all possible, can the authors to mention under each specific condition if there are potential cell-based T-reg therapies for that particular condition, that are currently either in the process of being approved or under clinical trials. This will be useful additional information; something to look forward to in the field of T-regs and autoimmunity. 
